# Visibility of noisy quantum dot-based measurements of Majorana qubits

Aleksei Khindanov[1], Dmitry I. Pikulin[2,3] and Torsten Karzig[3]

**1** Department of Physics, University of California, Santa Barbara, California 93106, USA
**2** Microsoft Quantum, Redmond, Washington 98052, USA
**3** Microsoft Quantum, Station Q, Santa Barbara, California 93106, USA

## Abstract

Measurement schemes of Majorana zero modes (MZMs) based on quantum dots (QDs) are of current interest as they provide a scalable platform for topological quantum computation. In a coupled qubit-QD setup we calculate the dependence of the charge of the QD and its differential capacitance on experimentally tunable parameters for both 2-MZM and 4-MZM measurements. We quantify the effect of noise on the measurement visibility by considering $1/f$ noise in detuning, tunneling amplitudes or phase. We find that on- or close-to-resonance measurements are generally preferable and predict, using conservative noise estimates, that noise coupling to the QDs is not a limitation to high-fidelity measurements of topological qubits.



# 1   Introduction

Majorana Zero Modes (MZMs) are explored as a promising platform for topological quantum computation [1–5]. As a direct consequence of their nonlocal nature, Majorana-based qubits are, in principle, less susceptible to decoherence and can provide better protected gates when compared to conventional qubits. Throughout the past decade a lot of experimental progress has been made on detecting signatures of the MZMs via observing zero-bias conductance peaks [6–12], a $4\pi$-periodic Josephson effect [13–15], signatures of exponential length-dependence of energy splittings [16, 17], and coherent single electron charge transfer between superconductors [18]. Though promising, these signatures have been proved inconclusive to make a definitive judgment on the presence of the MZMs in the system [19–26]. For this reason a measurement of a topological Majorana qubit draws significant attention from both experimental and theoretical standpoints. A successful implementation of such a readout of a topological qubit would mark the transition from studying properties of the topological phase to topologically protected quantum information processing. Moreover, as physically moving MZMs [27] currently appears to be practically challenging, measurement-based schemes [28, 29] come to the forefront as the most likely means of operating a Majorana-based topological quantum computer.

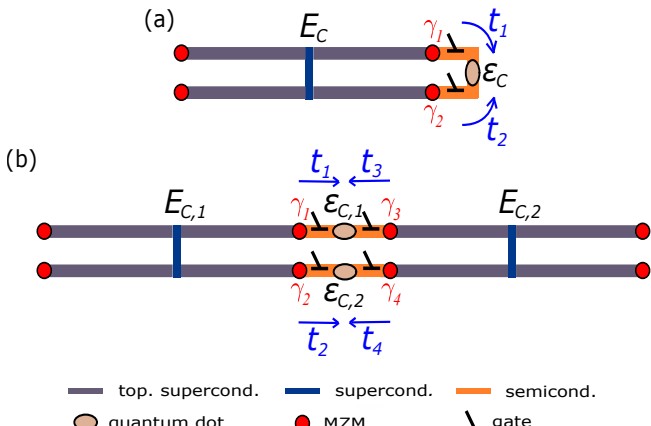

Figure 1: Schematic of the measurement setup of multi-MZM qubit islands. Only the measured MZMs of the qubit are labeled. (a) 2-MZM (single qubit) measurement setup. (b) 4-MZM (two qubit) measurement setup.

Various theoretical proposals for Majorana qubits and their readout procedure have been put forward [30–37]. Here we concentrate on the design for the qubit that features a superconducting island in the Coulomb blockaded regime [33, 34] consisting of two or more one-dimensional topological superconductors – realized for example in proximitized semiconductor nanowires [38, 39] – connected by a trivial superconductor. Each topological superconductor carries two MZMs at the ends. The qubit state is encoded by the parity of pairs of MZMs, e.g., $\sigma^z = i\gamma_i\gamma_j$, where $\sigma^z$ is Pauli operator in the computational space of the qubit and $\gamma_{i/j}$ are the corresponding Majorana operators. The total parity of a qubit island is conserved, which fixes the parity of the other two MZMs in 4-MZM islands. Measurements of the qubits are performed by coupling two (for single qubit measurements, see Fig. 1(a)) or four (for two-qubit measurements, see Fig. 1(b)) MZMs to quantum dots (QDs) while using parity-dependent shifts of the QD charge or capacitance as the readout signal. Such QD-based measurements are particularly promising since they can be embedded in scalable designs for the operation of topological qubits [34]. Motivated by this prospect, experimental studies of QD measurements in materials suitable for topological qubits are emerging [40, 41].

Despite the topological protection of Majorana qubits, quantum information storage and measurements are never perfect in practice due to sources of noise intrinsic and extrinsic to the qubit system. Quantifying the effect of noise is thus essential to understand the prospective performance of topological qubits. The effect of noise within the topological superconductors has been considered as the cause of the slow decoherence of idle qubits [42, 43] or as a possible reduction of the visibility of 2-MZM measurements [44]. Crucially, coupling the Majorana qubit to the QDs of the readout apparatus introduces new sources of noise. The desired effect of this noise is to collapse the qubit state into the outcome of the measurement [45, 46]. However, noise coupling to the QDs can also have negative effects on the visibility of the measurement and with the known susceptibility of QD to charge noise one might wonder whether QDs are a suitable platform for high-fidelity measurements. In this paper we study the effect of such noise on the measurement visibility and show that typical strengths of QD noise allow for high-fidelity qubit measurements.

To study the optimal operation point of measurements we will pay particular attention to the regime where the QD and the qubit island are tuned close to resonance (i.e. energy detuning between the two is much smaller than the MZM-QD coupling) in contrast to the widely applied far-detuned regime where the MZM-QD coupling is much smaller than the energy detuning and can be considered perturbatively [33, 34]. Such careful tuning to resonance can be particularly beneficial for 4-MZM measurements which were previously not discussed in this regime.

The rest of the paper is organized as follows. First, we review the single qubit measurements paying particular attention to the regime of the resonantly coupled island-QD system. We then extend this analysis to two-qubit measurements. Next, focusing on the single qubit measurement case, we analyze how noise in island-QD detuning affects the measurement visibility by calculating the signal-to-noise ratio (SNR) of the measurements. The Appendix presents details of calculations and treatment of the subleading noise sources – flux and coupling noise.

## 2 QD-based measurements

We start by reviewing how coupling a single QD to a pair of MZMs leads to a measurable change in the properties of the coupled MZM-QD system that depend on the parity of the MZMs before generalizing to measurements of four MZMs. As we show below, the regime of maximal measurement visibility is typically achieved when the QD and the qubit island

are tuned so that the energy configurations of an electron occupying the QD or the qubit island are close-to degeneracy. We therefore pay particular attention to this regime, which we refer to as resonant regime, and discuss how careful tuning enhances the visibility of 4-MZM measurements to be of similar order as the 2-MZM measurements.

## 2.1 2-MZM measurement

A typical setup for a 2-MZM single qubit measurement is depicted in Fig. 1(a). The effective low-energy Hamiltonian of the qubit-QD system is given by

$$\hat{H} = \hat{H}_{\text{C}} + \hat{H}_{\text{QD}} + \hat{H}_{\text{QD-MZM}}. \tag{1}$$

Here $\hat{H}_{\text{C}}$ is the charging energy Hamiltonian of the superconducting island, $\hat{H}_{\text{QD}}$ is a Hamiltonian of the QD and $\hat{H}_{\text{QD-MZM}}$ is a term describing tunneling between the island and the QD through MZMs.

Both $\hat{H}_{\text{C}}$ and $\hat{H}_{\text{QD}}$ contain charging energy contributions due to capacitance to the ground and between the subsystems. Additionally, $\hat{H}_{\text{QD}}$ contains the energy of the single-particle level on the QD. Due to charge conservation these contributions can be combined into:

$$\hat{H}_{\text{C+QD}} = \varepsilon_{\text{C}}(\hat{n} - n_{\text{g}})^2, \tag{2}$$

with $\hat{n}$ being the charge occupation of the QD while $\varepsilon_{\text{C}}$ and $n_{\text{g}}$ denote the effective charging energy and effective dimensionless gate voltage of the island-QD system. Expressions for the effective parameters in terms of original parameters of $\hat{H}_{\text{C}}$ and $\hat{H}_{\text{QD}}$ are given in Appendix A. Here we assumed a single-level QD without spin degeneracy, which is a valid assumption in high external magnetic field for small enough QD when the energy difference between the two lowest levels of the dot is larger than the MZM-QD coupling.

The tunneling Hamiltonian reads:

$$\hat{H}_{\text{QD-MZM}} = e^{-i\hat{\phi}}(t_1 f^{\dagger}\gamma_1 + t_2 f^{\dagger}\gamma_2) + \text{h.c.}, \tag{3}$$

where $t_{\alpha}$, $\alpha = 1, 2$ are coupling matrix elements of the MZMs to the fermionic mode on the QD described by creation operator $f^{\dagger}$. Note that since the Majorana operators are chargeless charge conservation is ensured by the operator $e^{i\hat{\phi}}$ that raises charge of the island by one electron charge. The couplings can be written as

$$t_{\alpha} = |t_{\alpha}|e^{i\phi_{\alpha}}; \ \alpha = 1, 2; \tag{4}$$

where the gauge invariant phase difference $\phi_1 - \phi_2$ depends on microscopic details of the matrix elements but can be tuned by varying the magnetic flux penetrating the enclosed area of the interference loop (see Fig. 1).

We now focus on the regime close to the QD-Majorana-island resonance, where $n_{\text{g}} = 1/2 + \Delta/2\varepsilon_{\text{C}}$ and the detuning $\Delta$ between the island and the QD level is $\Delta \ll \varepsilon_{\text{C}}$. The low energy Hamiltonian is then spanned by four states $|n, p\rangle$ where $n = 0, 1$ and $p = \pm 1$ are eigenvalues of the QD occupation and combined parity $p = p_{12}(-1)^n$ with $p_{12} = i\gamma_1\gamma_2$ being MZM parity. In contrast to the case of large detuning where the charge of the qubit island is fixed except for virtual tunneling events [33, 34] it is important to note that in the presence of the QD $p_{12}$ is no longer conserved. A non-demolition measurement therefore cannot directly determine $p_{12}$. Instead, the measurement outcome depends on the parity of the combined MZM-QD system $p$ which is a constant of motion in the absence of exponentially weak qubit dynamics [30, 45, 46]. Within the model discussed here, this manifests in a block-diagonal form of the Hamiltonian. Using the basis $|n, p\rangle$ with $|1, p\rangle = e^{-i\hat{\phi}}f^{\dagger}\gamma_1|0, p\rangle$ the elements of

the Hamiltonian blocks of given $p$ can be directly read off from Eqs.(2) and (3) with the parity dependence entering via $\langle 1, p | t_2 e^{-i\hat{\phi}} f^\dagger \gamma_2 | 0, p \rangle = -ipt_2$ such that

$$\hat{H}_p = \begin{pmatrix} \Delta/2 & \bar{t}_p^* \\ \bar{t}_p & -\Delta/2 \end{pmatrix}. \tag{5}$$

Here we introduced the effective MZM-QD coupling $\bar{t}_p = t_1 - ipt_2$. Equation (5) allows for a straight forward interpretation of the effect of $p$ on the MZM-QD system. Due to the interference of the two different paths that couple the QD and the qubit island, $p$ will control the strength of the effective coupling $|\bar{t}_p| = \sqrt{|t_1|^2 + |t_2|^2 + 2p|t_1 t_2| \sin\phi}$ where $\phi = \phi_2 - \phi_1$.

The parity-dependence of the coupling has measurable consequences for several observables and is used to diagnose the parity of the MZMs. The energy spectrum of the system takes the form

$$\varepsilon_{p,\pm} = \pm\frac{1}{2}\sqrt{\Delta^2 + 4|\bar{t}_p|^2}. \tag{6}$$

Figure 2(a) illustrates the energy spectrum in the case of $\phi = \pi/2$ and $|t_1| = 1.5|t_2|$. Even though optimal visibility is achieved when $|t_1| = |t_2|$ where $\bar{t}_p$ is either maximal or zero depending on the parity $p$, here we present plots away from this fine tuned point since a certain degree of the coupling asymmetry is expected in the QD-based readout experiments. Using the ground state of (6) the corresponding charge expectation value of the QD in the $\Delta \ll \varepsilon_C$ limit can be obtained as

$$\langle n_{\text{QD},p} \rangle = n_g - \frac{1}{2\varepsilon_C} \frac{\partial \varepsilon_{p,-}}{\partial n_g} = \frac{1}{2} + \frac{\Delta}{2\sqrt{\Delta^2 + 4|\bar{t}_p|^2}}. \tag{7}$$

The differential capacitance in the same limit takes the form

$$\frac{C_{\text{diff},p}}{C_g^2/C_{\Sigma,D}} = \frac{1}{2\varepsilon_C} \frac{\partial^2 \varepsilon_{p,-}}{\partial n_g^2} = -\frac{4\varepsilon_C |\bar{t}_p|^2}{(\Delta^2 + 4|\bar{t}_p|^2)^{3/2}}, \tag{8}$$

where $C_g$ is the capacitance between the gate and the QD and $C_{\Sigma,D} \equiv e^2/2\varepsilon_C$ is the total capacitance of the QD.

These two observables (7)-(8) can be measured in charge sensing, or quantum capacitance measurements respectively. Here we do not consider the details of the corresponding measurements but instead use the observables as a proxy for the measurement outcomes.

Fig. 2(b)-(c) depict the $\Delta$-dependence for various values of the phase $\phi$ of the charge expectation and differential capacitance for the ground state of the system at different parities $p$. In the absence of noise the parity dependence of the observables is strongest at $\phi = \pi/2$ and at or close to zero detuning.

In the most favorable regime close to zero detuning it becomes particularly important that $p$ is measured while we are ultimately interested in $p_{12}$ of the island decoupled from the QD. Failure to correctly infer $p_{12}$ from the measured value of $p$ would result in a measurement error and ultimately decrease readout and (in case of measurement-based topological quantum computing) gate fidelity. Connecting the measurement of $p$ to $p_{12}$ requires a well-defined initialization and finalization procedure of the measurement where the QD charge before and after the measurement is known. Charge conservation then allows to infer $p_{12}$ of the decoupled system from the measured $p$. A possible procedure is given by adiabatic tuning where the QD starts out and ends up far-detuned from resonance before and after the measurement to ensure a fixed charge state. The measurement is then initiated by first turning the MZM-QD coupling on and then tuning the system to resonance while the decoupling proceeds in opposite order. An alternative to this adiabatic tuning procedure would be to explicitly check the QD charge before and after the measurement by a separate charge measurement. Indeed, even if a close to adiabatic tuning is attempted such additional measurement might be required when one is aiming at very high measurement fidelities.

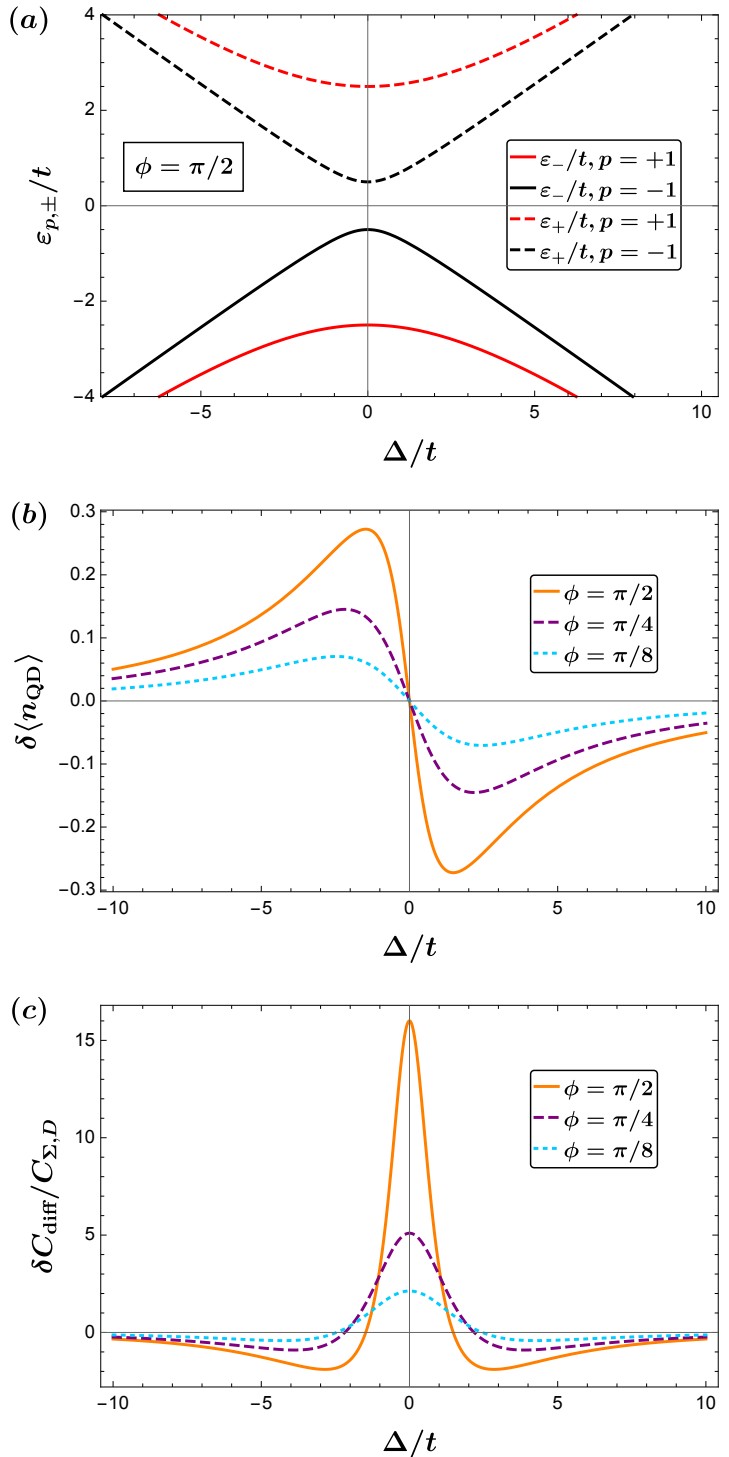

Figure 2: (a) MZM parity dependent part of the energies of the two lowest QD-MZM levels (6) as a function of island-QD detuning $\Delta$ in units of the MZM-QD hopping $t$. (b) Average QD charge difference between the two parity states $\delta\langle n_{\mathrm{QD}}\rangle = \langle n_{\mathrm{QD},p=+1}\rangle - \langle n_{\mathrm{QD},p=-1}\rangle$ as a function of detuning. (c) Differential capacitance difference between the two parity states $\delta C_{\mathrm{diff}} = C_{\mathrm{diff},+} - C_{\mathrm{diff},-}$ as a function detuning. We set $|t_1| = t$, $|t_2| = 1.5t$ for (a)-(c) and $C_g/C_{\Sigma,D} = 2$, $\varepsilon_C = 5t$ for (c).

## 2.2 4-MZM measurement

The setup for a 4-MZM measurement is shown in Fig. 1(b). 4-MZM measurements can be done utilizing only one QD. Here we consider two QDs since they provide greater tunability and are likely the generic case in scalable designs [34]. Similarly to 2-MZM situation, the effective low-energy Hamiltonian of this system has the form of (1). $\hat{H}_C$ and $\hat{H}_{QD}$ contributions are given in Appendix A while the tunneling Hamiltonian reads:

$$
\begin{aligned}
\hat{H}_{\text{QD-MZM}} = &e^{-i\hat{\phi}_1}(t_1 f_1^\dagger \gamma_1 + t_2 f_2^\dagger \gamma_2) \\
&+ e^{-i\hat{\phi}_2}(t_3 f_1^\dagger \gamma_3 + t_4 f_2^\dagger \gamma_4) + h.c.,
\end{aligned}
\tag{9}
$$

where $t_\alpha$ are couplings of the QDs described by fermionic operators $f_\beta^\dagger$ to the respective MZMs and $e^{i\hat{\phi}_\beta}$ is the raising operator of the charge of the island $\beta$.

For concreteness we consider the case where the system is tuned such that the lowest energy states are given by the 4 configurations of a single excess electron located on one of the QDs or islands. We denote the corresponding energies in the absence of tunnel coupling as $\varepsilon_\alpha$ with $\alpha \in \{i1, i2, d1, d2\}$ denoting the position of the electron. These energies are determined by the individual and mutual charging energies of the islands and QDs, and by the single-electron levels on the QDs. As in the 2-MZM case we will be particularly interested in the resonant regime where the energies $\varepsilon_\alpha$ become small. This requires tuning three parameters in general and can be done by tuning gate voltages on the two QDs and one of the islands.

Given that couplings of the low-energy subspace to MZMs other than $\gamma_1 \ldots \gamma_4$ are exponentially small, the total parity $p = p_{12}p_{34}(-1)^{n_1+n_2}$, where $n_\beta = f_\beta^\dagger f_\beta$, is conserved. We thus denote the low energy states as $|\alpha, p\rangle$:

$$
|i1, p\rangle = e^{-i\phi_1} e^{i\hat{\phi}_1} \gamma_1 f_1 |d1, p\rangle
\tag{10}
$$

$$
|i2, p\rangle = e^{-i\phi_3} e^{i\hat{\phi}_2} \gamma_3 f_1 |d1, p\rangle
\tag{11}
$$

$$
|d2, p\rangle = e^{i\phi_2} e^{-i\hat{\phi}_1} f_2^\dagger \gamma_2 |i1, p\rangle \,,
\tag{12}
$$

where we included the phases of the tunnel matrix elements $t_\alpha = |t_\alpha| e^{i\phi_\alpha}$ for convenience. In the above basis the Hamiltonian takes the form

$$
H = \begin{pmatrix}
\varepsilon_{d1} & |t_1| & |t_3| & 0 \\
|t_1| & \varepsilon_{i1} & 0 & |t_2| \\
|t_3| & 0 & \varepsilon_{i2} & -p|t_4|e^{i\phi} \\
0 & |t_2| & -p|t_4|e^{-i\phi} & \varepsilon_{d2}
\end{pmatrix},
\tag{13}
$$

where $\phi = \phi_1 - \phi_2 - \phi_3 + \phi_4$.

From the form of the Hamiltonian (13) it becomes clear that the energies of the system are independent on the individual 2-MZM parities and instead will depend via $\phi$ on the flux passing through the loop of the 4 tunneling junctions and on the overall parity $p$ which acts as a $\pi$ phase shift of $\phi$. Since the goal of the measurement is to ultimately determine 4-MZM parity $p_{12}p_{34}$ a similar tuning procedure as for 2-MZM measurements is required to fix the QD occupation. In fact, while the relation between $p$ and $p_{12}p_{34}$ suggests that the tuning procedure only needs to ensure that the joint QD parity $(-1)^{n_1+n_2}$ is the same before and after the measurement the charge occupation of all islands and QD need to remain unchanged by the measurement. The reason is that the measurement should determine $p_{12}p_{34}$ while not otherwise disturbing the quantum state of the qubits. Any net transfer of electrons between the islands or between the QDs relative to their state before the measurement would result in applying the corresponding operators involved in the electron transfer $\gamma_i \gamma_j$ to the qubit

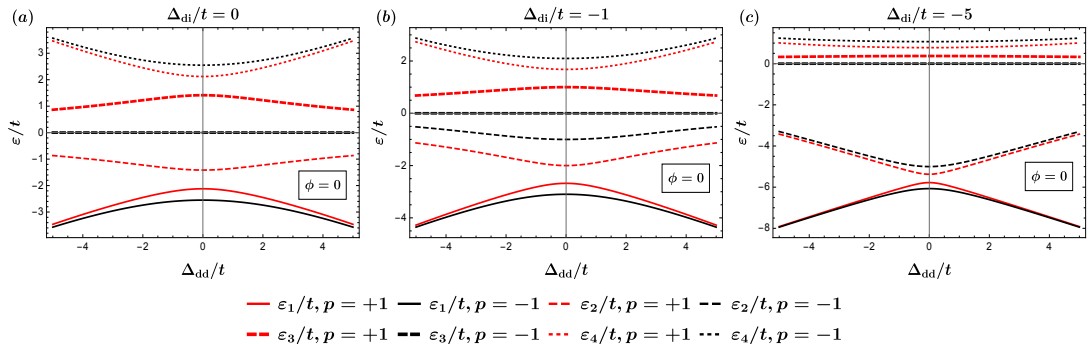

Figure 3: Eigenenergies of the Hamiltonian (13) for different parities $p$ and as a function of QD-QD detuning $\Delta_{dd}$ for various values of QD-island detuning $\Delta_{di}$. Here we set $|t_1| = |t_2| = 1.5t$, $|t_3| = |t_4| = t$. Panel (a) is given by the analytical expressions of Eq. (16).

states. Without a tuning procedure that ensures the occupation of the final configuration or an additional measurement to determine the configuration, the application of unknown pairs of Majorana operators would lead to dephasing. A possible tuning procedure from the resonant measurement configuration would work in a circular way: first detune the QD 1 to favor an occupation $n_1 = 1$, then tune island 1 to favor the empty state, followed by tuning QD 2 and island 2 to the empty state as well. Tuning all the couplings to zero then ensures a well defined charge configuration. The initialization procedure would be done in opposite order.

Exact diagonalization of (13) for arbitrary parameters involves cumbersome expressions. To gain intuition about the behavior of the energy levels we keep $\varepsilon_{i1} = \varepsilon_{i2}$ while introducing the QD detuning $\Delta_{dd} = \varepsilon_{d1} - \varepsilon_{d2}$ and the average detuning $\Delta_{di} = (\varepsilon_{d1} + \varepsilon_{d2})/2 - \varepsilon_{i1}$ between the QDs and islands. For now, we will also set $\Delta_{di} = 0$. The energy eigenvalues $\varepsilon$ are then given by the equation

$$\varepsilon^4 - \varepsilon^2 \left( \frac{1}{4} \Delta_{dd}^2 + t_\Sigma^2 \right) - \frac{1}{2} \varepsilon \Delta_{dd} t_\delta^2 + \bar{t}_p^{(4)}(\phi)^4 = 0, \tag{14}$$

in terms of $t_\Sigma^2 = \sum_{\alpha=1}^4 |t_\alpha|^2$, $t_\delta^2 = |t_1|^2 + |t_3|^2 - |t_2|^2 - |t_4|^2$ and the interference term

$$\bar{t}_p^{(4)}(\phi)^4 = |t_1 t_4|^2 + |t_2 t_3|^2 + 2p|t_1 t_2 t_3 t_4| \cos\phi. \tag{15}$$

The qualitative behavior is already captured by the case $t_\delta = 0$ which can be solved analytically yielding

$$\varepsilon_p^{(4)}(\phi) = \pm \frac{1}{\sqrt{2}} \sqrt{ \frac{\Delta_{dd}^2}{4} + t_\Sigma^2 \pm \sqrt{ \left( \frac{\Delta_{dd}^2}{4} + t_\Sigma^2 \right)^2 - 4\bar{t}_p^{(4)}(\phi)^4 }}. \tag{16}$$

The case of $t_\delta \neq 0$ is considered in Appendix C.

Going beyond $\Delta_{di} = 0$, Fig. 3 shows plots of the energy eigenvalues of (13) as functions of $\Delta_{dd}$ for $\phi = 0$, $t_\delta = 0$ and various values of island-QD detuning $\Delta_{di}$. For large negative $\Delta_{di}/t$ we recover the perturbative regime obtained in [34] where the parity dependent energy shift is of order $t^2/\Delta_{di}$. Figure 3 demonstrates that the energy differences between the ground states of different parity is maximal when the MZM-QD couplings are symmetric $|t_\alpha| = t$ and the system is on resonance $\Delta_{dd} = \Delta_{di} = 0$. By appropriately tuning the 4-MZM measurement system close to these parameters it becomes possible to reach a similarly strong parity dependence as in the case of 2-MZM measurements. Specifically, in the case of $\phi = 0$, $|t_\alpha| = t$, and $\Delta_{dd} = \Delta_{di} = 0$ one finds $\varepsilon_{+,gs}^{(4)} - \varepsilon_{-,gs}^{(4)} = (\sqrt{2} - 2)t$ which is of similar order as for the 2-MZM case. For a more explicit comparison of the capacitive response see App. B.

For the purpose of the following sections we note that the low energy part of the 4-MZM system spectrum in Fig. 3 (energies $\varepsilon_1$ and $\varepsilon_2$) qualitatively resembles the one of the 2-MZM system, see Fig. 2(a). Since all measurement visibility properties we consider in the next section are derived from the low energy part of the spectrum, we conclude that 4-MZM and 2-MZM cases are qualitatively similar in this regard and thus concentrate on the simpler 2-MZM case [1].

## 3 Noise and its effects on measurement visibility

We now describe the noise that will broaden the distribution of the observables. Here, we pay particular attention to intrinsic noise sources and their dependence on the system parameters. External noise sources, like amplifier noise, do not depend on the system parameters and are uncorrelated with the system noise – therefore they can be added straight-forwardly. The leading internal noise source in the measurement setup of Fig. 1 would likely be the charge noise which affects the on-site energy and thus detuning of the QDs. In our study we assume the $1/f$ power spectrum of the charge noise which has been reported in other QD-based devices, most notably semiconductor charge qubits [47–49]. We discuss noise in the strength of the tunnel couplings and flux noise which affects the phase $\phi$ in Appendices G and H. Using noise estimates from related experimental setups we conclude that these noise sources likely play a subleading role on the visibility of the measurement compared to the charge noise considered in the main text.

We first formulate the general framework of how we treat noise. Consider an observable $\hat{y}(x(t))$ that depends on the parameter $x(t) = x + \delta x(t)$, where $x$ is the fixed setting of the parameter $x$ and $\delta x(t)$ is the time-dependent noise. We describe the noise perturbatively by considering the second order expansion in the parameter of noise:

$$\hat{y}(x(t)) = \hat{y}_0(x) + \hat{y}_1(x)\delta x(t) + \frac{1}{2}\hat{y}_2(x)\delta x(t)^2, \tag{17}$$

where $\hat{y}_0$ is the unperturbed observable and $\hat{y}_1$, $\hat{y}_2$ are first and second derivatives of $\hat{y}_0$ with respect to $x$. Since measurements are recorded over a finite measurement time $\tau_{\mathrm{m}}$ we are ultimately interested in the time averaged quantities $\hat{Y} = \frac{1}{\tau_{\mathrm{m}}} \int_0^{\tau_{\mathrm{m}}} dt\, \hat{y}(x(t))$. We use the expectation value $Y = \langle \hat{Y} \rangle$ and variance $\sigma_Y^2 = \langle \hat{Y}^2 \rangle - \langle \hat{Y} \rangle^2$ to determine the measurable signal and internal noise.

The above expectation value $\langle \dots \rangle$ is taken with respect to the environment for the noisy parameter. There are two opposing limits how to incorporate a finite temperature in the expectation values of the system operators. (1) The operator $\hat{Y}$ is temperature independent and the expectation value is taken with respect to the full density matrix of the system which includes both finite-temperature and noise effects; (2) the operator $\hat{Y}$ is already the temperature-averaged observable (i.e. the expectation value with respect to the unperturbed finite-temperature density matrix has been already taken) in which case taking the expectation value $\langle \dots \rangle$ amounts to only performing noise-averaging. Method (1) would give a finite variance even in the absence of noise due to temperature fluctuations while (2) only includes fluctuations due to noise. These differences only become important for temperatures that allow excitations above the ground state. In the case when there is a significant occupation of the excited state, the timescales involved in the temperature fluctuations determine which

---

[1]Technically, the 4-MZM measurement is performed by measuring charge/capacitance of one of the dots and in order to compare it to the 2-MZM case, one needs to plot the spectra of Fig. 3 as functions of variables $\varepsilon_{d1}, \varepsilon_{d2}$ rather than $\Delta_{dd}, \Delta_{di}$. However, the two variable sets are related to each other by simple linear transformation and thus the spectra as a function of, for example, $\varepsilon_{d1}$ looks rotated with respect to the ones in Fig. 3 such that the ground state part still qualitatively resembles the one in Fig. 2(a)

$$Y(p = +1) \quad \rule{3cm}{1.5pt} \quad \updownarrow 2\sigma_Y(p = +1)$$

$$Y(p = -1) \quad \rule{3cm}{1.5pt} \quad \updownarrow 2\sigma_Y(p = -1)$$

Figure 4: Diagram explaining definition of the signal and noise given by Eqs. (20),(21). $Y$ is a measured quantity which depends on $p$, black lines indicate respective values of $Y$. The red line is a level broadening due to the noise with a standard deviation $\sigma_Y$.

of the two methods are more appropriate in capturing the variance of the measurement outcomes. If during the measurement time the system transitions frequently between the ground and excited state, the measurement will probe temperature averaged quantities (2), while for transitions slower than the measurement time the distribution of measurement outcomes would be broadened by temperature (1). To focus on the effect of the noise we take the limit (2) of long measurement times.

We assume that the expectation values of the fluctuations are fully described by the spectral function $S_x(\omega)$ of the noise via $\langle \delta x(0) \delta x(t) \rangle = \int d\omega e^{i\omega t} S_x(\omega)$. For the $1/f$ noise which we assume below $S_x(\omega) = \alpha_x/|\omega|$ we find up to second order in the noise

$$Y = y_0 + y_2 \alpha_x \left( 1 - \gamma - \log(\omega_{\min} \tau_c / 2) \right) \tag{18}$$

$$\sigma_Y^2 = y_1^2 \alpha_x c + \frac{y_2^2}{2} \alpha_x^2 \left( 5 + c^2 \right), \tag{19}$$

where $\gamma \approx 0.577$ is Euler's constant and $c = 3 - 2\gamma - 2\log(\omega_{\min} \tau_m)$, see Appendix D for details. Note that the nature of $1/f$ noise requires to introduce low and high frequency cutoffs of the noise in addition to the finite measurement time. The cutoffs can be physically motivated. The high frequency cutoff arises due to finite correlation time of the noise. For short times $t \ll \tau_c$ one expects $\langle \delta x(0) \delta x(t) \rangle$ to approach a constant. The specific value of this time scale is not important due to the weak logarithmic dependence. We associate $\tau_c^{-1}$ with the highest frequency that the measurement apparatus can possibly resolve. Noise at higher frequencies simply averages out and cannot be detected during measurement. The measurements are performed by coupling resonators to the quantum dot and observing shifts in the resonance frequency. This frequency thus provides a natural cutoff for the time scale the detector can resolve. Typical resonator frequencies are $\sim 1$ GHz and thus we set $\tau_c^{-1} = 1$ GHz. The low frequency cutoff $\omega_{\min}$ is given by the inverse timescale at which the system is recalibrated since very slow components of the noise act as drift which can be removed by calibration.

While the dependence on $\omega_{\min}$ is weak it should be noted that Eqs. (18),(19) emphasize that similar to conventional qubits, the measurement apparatus of topological qubits needs to be regularly recalibrated. For numerical estimates we use $\omega_{\min}^{-1} = 10\tau_m$ with $\tau_m = 1$ $\mu$s. For longer recalibration times the noise will grow slowly as $\sqrt{\log(\omega_{\min} \tau_m)}$. This effect becomes relevant when considering very long time-intervals between calibration that might be desirable for quantum computation. For example, for $\omega_{\min}^{-1} \sim 1$ day with $\tau_m = 1$ $\mu$s the noise would be increased by a factor $\sim 3$ relative to our estimates.

During the qubit readout the goal is to be able to differentiate between parity $p = +1$ and $p = -1$ states by measuring the observables discussed in the previous section. This is schematically illustrated in Fig. 4. Thus, for the particular case of measurement visibility

analysis, we define the signal $\mathcal{S}$ and the noise $\mathcal{N}$ in variable $Y$ as

$$\mathcal{S}_Y = |Y(p = +1) - Y(p = -1)| \tag{20}$$

$$\mathcal{N}_Y = \sigma_Y(p = +1) + \sigma_Y(p = -1). \tag{21}$$

# 4 Detuning noise

The dominant source of noise in the island-QD detuning $\Delta$ is the gate voltage noise on the QD $n_g$ which is typically dominated by $1/f$ charge noise

$$S_\Delta(\omega) = \varepsilon_C^2 \frac{\alpha_C}{|\omega|}. \tag{22}$$

Here we explicitly wrote the coupling strength of the noise to the system which is controlled by the charging energy $\varepsilon_C$ and the strength of the noise described by the dimensionless parameter $\alpha_C$ that depends on the environment that is causing the charge noise. The latter depends on the experimental setup and materials.

We estimate $\alpha_C$ by considering the strength of dephasing of charge qubits in InAs/Al hybrid systems that are investigated for their potential use as building blocks for Majorana qubits. Reference [18] reports coherence time of the InAs/Al based superconducting charge qubit with $\varepsilon_C/h \sim 10$ GHz to be $T_2^* \sim 1$ ns. A simple estimate for the dephasing caused by charge noise is given by $T_2^* \sim \hbar/(\varepsilon_C \sqrt{\alpha_C})$ [42]. We use this relation to estimate the experimentally relevant $\sqrt{\alpha_C} \sim 0.01$. Typical values for the charging energy of InAs QDs are $\varepsilon_C \sim 100\mu eV$ [40, 41] which leads to $\varepsilon_C \sqrt{\alpha_C} \sim 1\mu eV$. Note that this gives a conservative estimate for the strength of charge noise for the topological qubit as it assumes no optimization of noise as compared to current experimental capabilities. Similar estimates for charge qubits in much more mature GaAs-based systems yield $\sqrt{\alpha} \sim 10^{-4}$ [42, 48]. The perturbative treatment of the noise in Eq. (17) close to $\Delta = 0$ is justified for the charge noise as long as $\sqrt{\alpha_C}\varepsilon_C \ll |\bar{t}_p|$. The above estimate of $\sqrt{\alpha_C} \ll 1$ therefore justifies the perturbative treatment as long as the effective tunnel couplings $\bar{t}_p$ are not too small compared to $\varepsilon_C$. For our numerical estimates we take $|t_1| = t$, $|t_2| = 1.5t$, $t = \varepsilon_C/5$ which guarantees validity of the perturbative treatment for the entire parameter range of $\Delta$ and $\phi$. The coupling asymmetry $|t_2|/|t_1| = 1.5$ does not correspond to the case of maximum visibility (which is reached for $|t_2|/|t_1| = 1$). However, a certain degree of the coupling asymmetry is expected in the QD-based readout experiments as the coupling fine-tuning might pose a challenge.

Using expressions of Eq. (7) (Eq. (8)) for the average QD charge (differential capacitance of the QD) in the 2-MZM measurement case we plot the zero-temperature dependence of the signal $\mathcal{S}_n (\mathcal{S}_C)$ and noise $\mathcal{N}_n (\mathcal{N}_C)$ given by Eqs. (20) and (21) in terms of the phase and detuning in Figs. 5 and 6. Temperature dependence of detuning noise is analyzed in Appendix E.

The dependence on the detuning $\Delta$ shows that the charge signal $\mathcal{S}_n$ takes its maximal value for $\Delta = \Delta_n^{\max}$ with $\Delta_n^{\max} \sim t$. This follows from the suppression of the signal at $\Delta = 0$ ($\Delta \to \infty$) due to the QD charge reaching $\langle n_{QD} \rangle = 1/2$ ($\langle n_{QD} \rangle = 1$) independent of parity. Neglecting noise one can find analytically $\Delta_n^{\max} = 2|\bar{t}_-^2 \bar{t}_+|^{1/3}/\sqrt{1 + |\bar{t}_-/\bar{t}_+|^{2/3}}$. We checked numerically that for our choice of parameters the noise-induced term in $\mathcal{S}_n$ is perturbative, i.e. much smaller then the noise-free term, and thus produces small corrections to this analytical result.

In the regime of perturbative noise and $T = 0$ the differential capacitance signal $\mathcal{S}_C$ is always maximal at $\Delta = 0$ while vanishing at $\Delta = \Delta_C^{\min} = \Delta_n^{\max}$. The latter marks the point where the differential capacitance corresponding to the smaller of $|\bar{t}_-|$ and $|\bar{t}_+|$, which is generally dominating around small detuning due to a larger curvature, is equal to the differential capacitance of the larger coupling which dominates in the regime of large detuning.

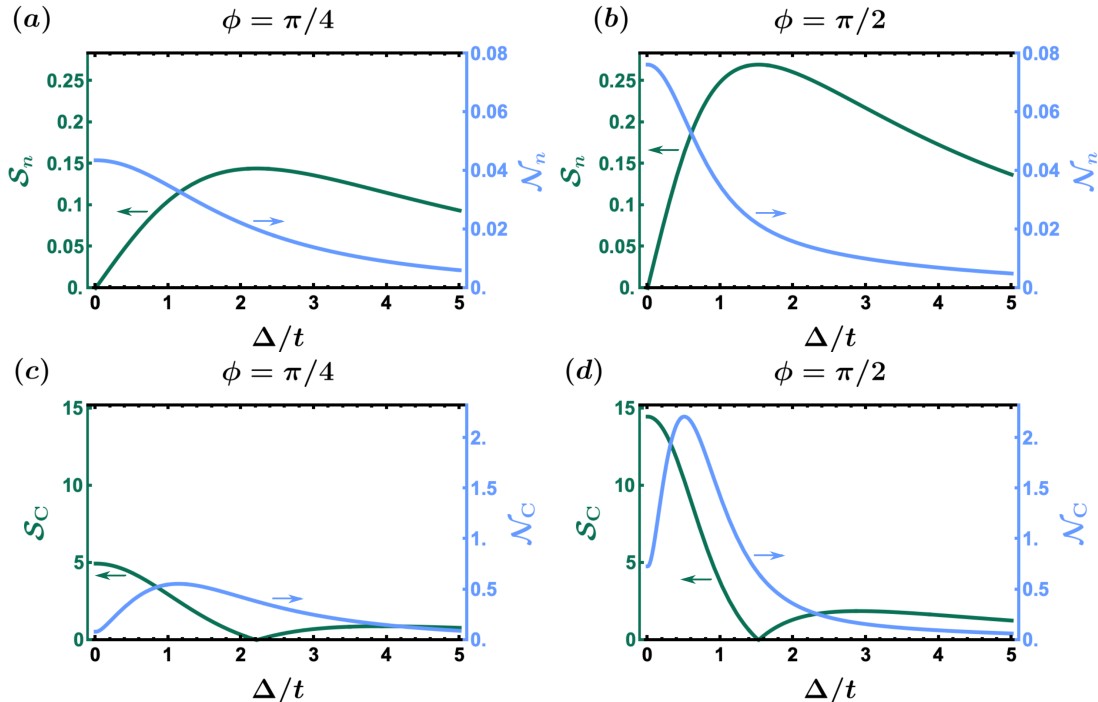

Figure 5: Signal (20) and noise (21) for the 2-MZM measurements of the average QD charge $\langle n_{\mathrm{QD}} \rangle$ (a)-(b) and differential QD capacitance $C_{\mathrm{diff}}/C_{\Sigma,\mathrm{D}}$ (c)-(d) as a function of detuning $\Delta$ for different values of $\phi$. Here we assume that the system is in its ground state ($T = 0$) and set $|t_1| = t$, $|t_2| = 1.5t$, $t = \varepsilon_{\mathrm{C}}/5 = 0.02$ meV, $C_g/C_{\Sigma,\mathrm{D}} = 2$, and the noise is detuning noise of strength $\sqrt{\alpha_C} = 0.01$.

Note that at finite temperature and in the presence of noise $\Delta = 0$ might not always be the point of maximal signal and $\Delta_C^{\min}$ might differ from $\Delta_n^{\max}$. Consider for example the regime of extreme fine tuning where $|\bar{t}_-| \ll T, \sigma_\Delta$ while $|\bar{t}_+| \gg T, \sigma_\Delta$. In that limit the contribution to the differential capacitance of the $p = -1$ parity would vanish as $C_{\mathrm{diff},-} \propto |\bar{t}_-|/(T\sigma_\Delta)$. A derivation of this expression is given in Appendix F. Approaching this regime would mean that $\Delta_C^{\min}$ would shift to values smaller than $\Delta_n^{\max}$ and eventually reach $\Delta_C^{\min} = 0$. Further reducing $|\bar{t}_-|$ would make $\mathcal{S}_C$ be dominated by the $p = +1$ branch independent of $\Delta$ thus restoring $\Delta = 0$ as the point of maximal signal. Naturally, the limit of very small $|\bar{t}_-|$ breaks the perturbative treatment of noise used in this paper. Nevertheless, as long as the noise $\sigma_\Delta$ is weak compared to $|\bar{t}_+|$ results for the limit $|\bar{t}_-| \to 0$ can be obtained using our formalism by replacing $\mathcal{S}_C \to C_{\mathrm{diff},+}$ and $\mathcal{S}_n \to \langle n_{\mathrm{QD},+} \rangle - \langle n_\sigma \rangle$, where $\langle n_\sigma \rangle$ is charge expectation value for vanishing coupling broadened by noise [2].

The noise $\mathcal{N}_n$ is maximal at $\Delta = 0$ and falls off quickly for large detuning. From the perspective of pure charge noise the SNR would thus be largest for large detuning where the signal is also becoming suppressed. The presence of other noise sources will limit this behavior. For example the effect of external amplifier noise is typically minimized for the strongest signal. At the maximal signal, i.e. $\Delta = \Delta_n^{\max}$, we find a charge-noise-limited SNR of $\approx 12$ for $\phi = \pi/2$. Thus, as long as the integration times are not sufficiently long to extend the amplifier-limited SNR beyond 12 the point of maximal experimental SNR will be close to $\Delta = \Delta_n^{\max}$.

The noise $\mathcal{N}_C$ shows a local minima at $\Delta = 0$ due to the absence of the first-order con-

---

[2]For Gaussian noise of width $\sigma_\Delta$ the charge of the ground ($-$) and excited ($+$) state is broadened via $n_{\sigma,\pm} = (1 \mp \mathrm{erf}(\Delta/\sqrt{2}\sigma_\Delta))/2$. The expectation value $\langle n_\sigma \rangle$ is then given by appropriately temperature averaging the ground and excited state contribution.

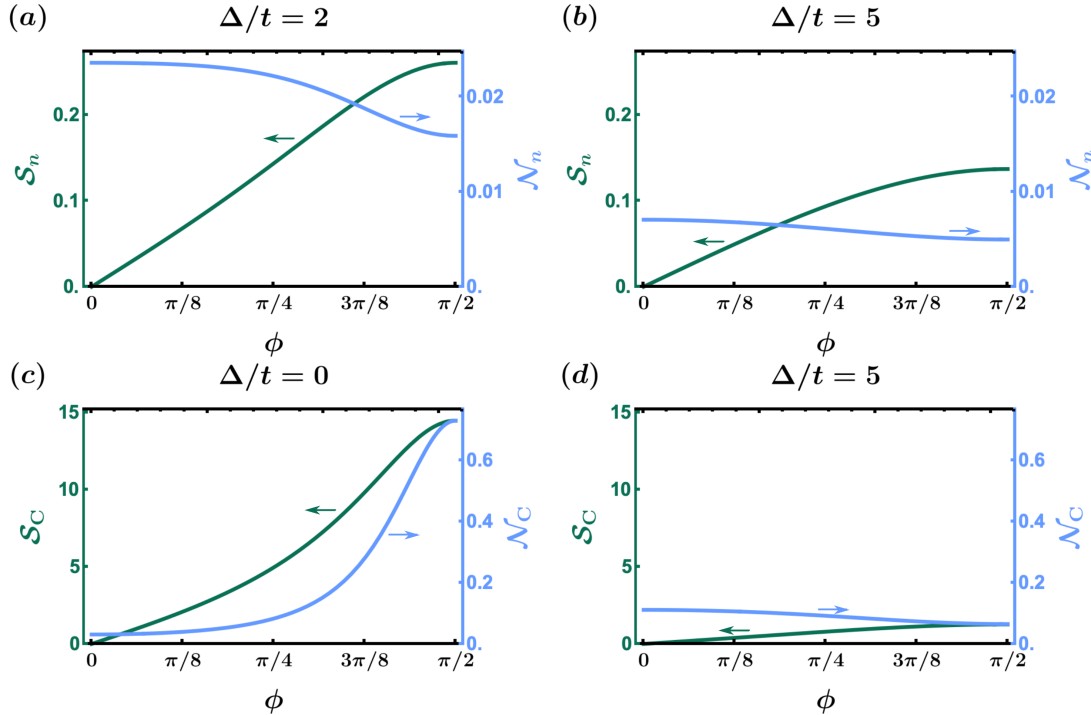

Figure 6: Signal (20) and noise (21) for the 2-MZM measurement of the average QD charge $\langle n_{\mathrm{QD}} \rangle$ (a)-(b) and differential QD capacitance $C_{\mathrm{diff}}/C_{\Sigma,\mathrm{D}}$ (c)-(d) as a function of phase $\phi$ for different values of $\Delta/t$. We used the same parameters as in Fig. 5.

tribution of charge noise. This emphasizes that for capacitive measurements $\Delta = 0$ is likely the optimal operation point. The only exception is the above-mentioned regime where the smaller of the effective couplings, say $|\bar{t}_-|$, is accidentally of the order of $T\sigma_\Delta/|\bar{t}_+|$. For the parameters we used, we find a charge-noise-limited SNR of $\approx 20$ for $\phi = \pi/2$.

Figure 6 shows the $\phi$-dependence of the signal and noise. The main effect of changing $\phi$ is to increase the difference between $|\bar{t}_+|$ and $|\bar{t}_-|$ as $\phi$ approaches $\pi/2$. This generally increases the signal for all observables as long as the noise remains perturbative. Away from $\Delta = 0$ changing $\phi$ has only a relatively weak effect on the noise which means that for charge measurements $\phi \to \pi/2$ is always preferable. In the case of capacitance measurements that are operated at $\Delta = 0$ approaching $\phi = \pi/2$ not only increases the signal but also the noise. The optimal SNR can thus be obtained away from $\phi = \pi/2$. Similar to the discussion of the effect of noise for the charge measurements at large detuning, external constraints will determine whether the increase in the signal or the reduction of the noise are more important for obtaining the best experimental SNR.

## 5 Conclusion

In the present work we identified detuning charge noise as the dominant source of intrinsic noise that affects the measurement visibility of Majorana qubits probed by QDs. We studied the Hamiltonian for 2-MZM and 4-MZM measurements non-perturbatively in the tunnel coupling and emphasized the similarity of their description in the regime of small detunings which in general optimizes SNR in the presence of external noise. 4-MZM measurements require more tuning and more manipulations to bring the system into the optimal measurement regime, but can produce signal of the *same* order as 2-MZM measurements. We thus analyzed the 2-MZM

measurement for SNR in detail and claim the 4-MZM one will behave similarly.

Generally we obtain large SNRs $\gtrsim 10$ for conservative assumptions on the charge noise of the system that is tuned to the optimal measurement regime. Since we did not explicitly treat external noise sources like amplifiers our SNRs should be understood as the limiting SNRs that can be obtained after long measurement times. The large obtained SNRs indicate that charge noise will likely not be limiting the fidelity of measurements of topological qubits.

We make concrete predictions for the visibility of the topological qubit measurement, but our results are relevant for and can be tested in simpler setups. For example, test devices replacing the qubit island with another QD show similar interference effects. Our SNRs can then be understood as describing the difference between measurements where the enclosed phases of the tunnel couplings are $\phi$ and $\phi + \pi$.

*Note added.* Recently a related manuscript appeared addressing the effects of the charge noise on 2-MZM measurements [50]. The authors treat noise in the detuning non-perturbatively by convolving the signal by a phenomenological Gaussian broadening. The explicit treatment of the $1/f$ character of the charge noise presented here could be used to better inform the parameters of the Gaussian broadening.

# Acknowledgements

We thank Roman Lutchyn and Bela Bauer for useful discussions.

# A  Island(s) and QD(s) contributions to the total low-energy Hamiltonian of the qubit(s)-QD(s) system

## A.1  2-MZM case

In this Appendix we derive effective Hamiltonian $\hat{H}_{\text{C+QD}}$ of Eq. (2). The island and QD contributions to the total low-energy Hamiltonian (1) of the coupled island-QD system take the form $\hat{H}_{\text{C}} = E_{\text{C}}(\hat{N} - N_g)^2 + E_{\text{M}}(\hat{N} - N_g)(\hat{n} - n_g)$ and $\hat{H}_{\text{QD}} = h\hat{n} + \varepsilon_{\text{C}}(\hat{n} - n_g)^2$ respectively, where $\hat{N}(\hat{n})$ is a charge occupation of the island(QD), $E_{\text{C}}(\varepsilon_{\text{C}})$ is a charging energy of the island(QD), $N_g(n_g)$ is a dimensionless gate voltage of the island(QD), $E_{\text{M}}$ is a mutual charging energy between the island and the QD and $h$ is energy of a single electron level of the QD. Here we assumed a single-level QD without spin degeneracy, which is a valid assumption in high external magnetic field for small enough QD.

Total charge conservation in the system dictates that $\hat{N} + \hat{n} = N_{\text{tot}}$, where $N_{\text{tot}}$ is a total number of electrons in the system. Using this, $\hat{H}_{\text{C}} + \hat{H}_{\text{QD}}$ can be rewritten in terms of only one operator, e.g. $\hat{n}$, yielding up to a constant term:

$$\hat{H}_{\text{C}} + \hat{H}_{\text{QD}} = (E_{\text{C}} + \varepsilon_{\text{C}} - E_{\text{M}})(\hat{n} - \tilde{n}_g)^2 \,, \tag{23}$$

with effective dimensionless gate voltage given by

$$\tilde{n}_g = \frac{E_{\text{C}}(N_{\text{tot}} - N_g) + \varepsilon_{\text{C}} n_g - h/2 - E_{\text{M}}(N_{\text{tot}} - N_g + n_g)/2}{E_{\text{C}} + \varepsilon_{\text{C}} - E_{\text{M}}} \,. \tag{24}$$

Relabeling parameters in terms of effective ones as $E_{\text{C}} + \varepsilon_{\text{C}} - E_{\text{M}} \to \varepsilon_{\text{C}}$, $\tilde{n}_g \to n_g$, we get $\hat{H}_{\text{C}} + \hat{H}_{\text{QD}} \to \hat{H}_{\text{C+QD}}$ with $\hat{H}_{\text{C+QD}}$ given by Eq. (2).

### A.2 4-MZM case

Denoting $\hat{N}_i(\hat{n}_i)$, $i = 1,2$ as a charge occupation of the $i$th island(QD), $N_{g,i}(n_{g,i})$, $i = 1,2$ as a dimensionless gate voltage of the $i$th island(QD), $h_i$, $i = 1,2$ as energy of a single electron level of the $i$th QD, we describe the islands and QDs contribution to the low-energy Hamiltonian of the 4-MZM system as

$$\hat{H}_{\mathrm{C}} + \hat{H}_{\mathrm{QD}} = \frac{e^2}{2} \sum_{i,j=1}^{4} \hat{v}_i P_{ij} \hat{v}_j + \sum_{i=1,2} h_i \hat{n}_i \,, \tag{25}$$

up to a constant. Here $\hat{v}_i$ represent dimensionless excess charge of the islands/QDs, i.e.

$$\hat{v}_i = \begin{cases} (\hat{N}_i - N_{g,i}), \ i = 1,2 \\ (\hat{n}_{i-2} - n_{g,i-2}), \ i = 3,4 \end{cases} \tag{26}$$

and $P_{ij}$ are matrix elements of the inverse of the $4 \times 4$ capacitance matrix of the system with the first(last) two indices representing the islands(QDs) degrees of freedom. The first term in Eq. (25) corresponds to the total electrostatic energy of the system while the second term is a total orbital energy of the QD levels. Notice that the Hamiltonian is always diagonal in the basis of the occupation numbers of islands and quantum dots. Thus, energies $\varepsilon_\alpha$, $\alpha \in \{i1, i2, d1, d2\}$ introduced in Section 2.2 can be found by plugging appropriate values $\hat{n}_i, \hat{N}_i \in \{0,1\}$ of the island and QD charges into Eq. (25).

## B  Quantitative comparison of capacitance of 2- and 4-MZM measurements

To gain intuition about the different signal strengths of the 2- and 4-MZM measurements we compare the corresponding curvatures $\partial^2 \varepsilon / \partial \Delta^2$ of the two cases which are proportional to the capacitive response $C_{\mathrm{diff},p}$. In particular we consider the case $\Delta_{\mathrm{dd}} = \Delta = 0$ using Eqs. (6) and (16) since finite $\Delta_{\mathrm{dd}}$ expressions are too complicated in the 4-MZM case to be illuminating. We find:

$$\left.\frac{\partial^2}{\partial \Delta^2}\right|_{\Delta=0} \varepsilon_{p,\mathrm{gs}}(\phi) \;=\; -\frac{1}{4|\bar{t}_p(\phi)|} \tag{27}$$

$$\left.\frac{\partial^2}{\partial \Delta_{\mathrm{dd}}^2}\right|_{\Delta_{\mathrm{dd}}=0} \varepsilon_{p,\mathrm{gs}}^{(4)}(\phi) \;=\; -\frac{\sqrt{t_\Sigma^2 + \sqrt{t_\Sigma^4 - 4\bar{t}_p^{(4)}(\phi)^4}}}{4\sqrt{2}\sqrt{t_\Sigma^4 - 4\bar{t}_p^{(4)}}}\,. \tag{28}$$

Note that $t_\Sigma^4 \geq 4\bar{t}_p^{(4)}$, we therefore see that $\sqrt{t_\Sigma^4 - 4\bar{t}_p^{(4)}(\phi)^4}/t_\Sigma$ plays a similar role for the 4-MZM case as $|\bar{t}_p(\phi)|$ does for the 2-MZM case.

As mentioned in the main text, for perfectly symmetric tuning $|t_\alpha| = t$, the 2- and 4-MZM cases show parity-dependent energy shifts of similar order. We now look at the capacitive responses in the same limit. Let's consider $p = 1$ ($p = -1$ can be obtained by shifting $\phi \to \phi + \pi$). We then find,

$$\left.\frac{\partial^2}{\partial \Delta^2}\right|_{\Delta=0} \varepsilon_{+,\mathrm{gs}}(\phi - \pi/2) \;=\; -\frac{1}{8t} \frac{1}{\sin(\phi/2)} \tag{29}$$

$$\left.\frac{\partial^2}{\partial \Delta_{\mathrm{dd}}^2}\right|_{\Delta_{\mathrm{dd}}=0} \varepsilon_{+,\mathrm{gs}}^{(4)}(\phi) \;=\; -\frac{1}{8t} \frac{\cos(\phi/4 - \pi/4)}{\sin(\phi/2)}\,, \tag{30}$$

where we used simplifications that apply without loss of generality for $0 \leq \phi \leq 2\pi$ which bounds $1/\sqrt{2} \leq \cos(\phi/4 - \pi/4) \leq 1$. We therfore see that in this limit the 2- and 4-MZM capacitive response behaves very similarly differing by at most a factor of $\sqrt{2}$.

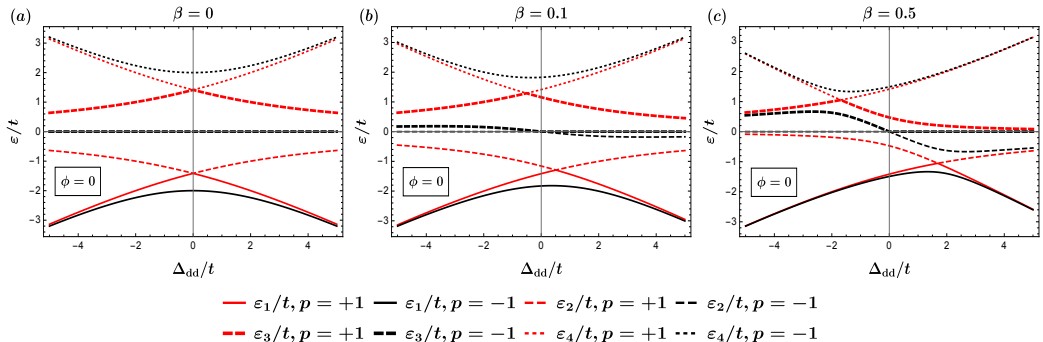

Figure 7: Eigenenergies of the Hamiltonian (13) for different parities $p$ and as a function of QD-QD detuning $\Delta_{\mathrm{dd}}$ for various values of coupling asymmetry $\beta$. Here we set $|t_1| = |t_3| = t$, $|t_2| = |t_4| = t(1-\beta)/(1+\beta)$, $\Delta_{\mathrm{di}} = 0$. Panel (a) is given by the analytical expressions of Eq. (16).

## C   4-MZM measurement in case $t_\delta \neq 0$

In case of $t_\delta \neq 0$ solutions of the quartic equation (14) have cumbersome analytical form and we do not present them here. Instead, we parametrize the coupling asymmetry giving rise to finite $t_\delta$ using the parameter $\beta$ such that $|t_1| = |t_3| = t$, $|t_2| = |t_4| = t(1-\beta)/(1+\beta)$, $t_\delta^2 = 8\beta t^2/(1+\beta)^2$ and plot solutions of Eq. (14) as functions of $\Delta_{\mathrm{dd}}$ in Fig. 7 for $\phi = 0$ and various values of $\beta$. Fig. 7 illustrates that finite values of $t_\delta$ introduce a shift of the crossings (or avoided crossings if $|t_1| \neq |t_3|$ and/or $|t_2| \neq |t_4|$) away from $\Delta_{\mathrm{dd}} = 0$ which can be calculated analytically giving

$$\Delta_{\mathrm{dd}}^{\mathrm{shift}} = \pm\sqrt{2}t_\delta^2 \sqrt{\frac{t_\Sigma^2 - \sqrt{t_\Sigma^4 - 4\bar{t}_p^{(4)}(\phi)^4 - t_\delta^4}}{4\bar{t}_p^{(4)}(\phi)^4 + t_\delta^4}}. \tag{31}$$

Taking into account this shift in $\Delta_{\mathrm{dd}}$, the ground state part of the 4-MZM system spectrum in case the of $t_\delta \neq 0$ still looks qualitatively similar to the one of the 2-MZM system spectrum depicted in Fig. 2(a) for all cases shown in Fig. 7.

## D   SNR for $1/f$ noise

In this appendix we derive expressions for $Y$ and $\sigma_Y$ for Gaussian noise that is fully described by a two-point correlation function with $1/f$ spectral power density $S_x(\omega) = \alpha_x/|\omega|$. Using the expansion of Eq. (17) together with $\langle\delta x\rangle = 0$ we find

$$Y = y_0 + \frac{y_2}{2\tau_{\mathrm{m}}} \int_0^{\tau_{\mathrm{m}}} dt \langle\delta x(t)^2\rangle = y_0 + \frac{y_2}{2} \int d\omega S_x(\omega). \tag{32}$$

Strictly speaking this expression is divergent but physical constraints provide frequency cutoffs for $S(\omega)$. The low frequency cutoff $\omega_{\mathrm{min}}$ is given by the time that passed since the measurement apparatus was calibrated. Calibration redefines very slow noise components into the signal. In general $\omega_{\mathrm{min}}^{-1} > \tau_{\mathrm{m}}$ but depending on the way the qubit is operated $\omega_{\mathrm{min}}^{-1}$ could exceed $\tau_{\mathrm{m}}$ by several orders of magnitude. The high frequency cutoff is given by the inverse of the correlation time of the noise $\tau_{\mathrm{c}}$ since for $t' < \tau_{\mathrm{c}}$ one would expect $\langle\delta x(0)\delta x(0)\rangle \approx \langle\delta x(0)\delta x(t')\rangle$.

We thus regularize Eq. (32) via

$$Y = y_0 + y_2 \alpha_x \int_{\omega_{\min}}^{\infty} d\omega \frac{1}{\omega} \frac{1}{\tau_c} \int_{-\tau_c/2}^{\tau_c/2} dt e^{i\omega t} \approx y_0 + y_2 \alpha_x \big(1 - \gamma - \log(\omega_{\min} \tau_c/2)\big), \quad (33)$$

where $\gamma \approx 0.577$ is the Euler's constant and we used that $\omega_{\min} \tau_c \ll 1$.

The variance is given by

$$\sigma_Y^2 = \frac{1}{\tau_m^2} \int_0^{\tau_m} \int_0^{\tau_m} dt dt'$$
$$\left\{ y_1^2 \langle \delta x(t) \delta x(t') \rangle + \frac{y_2^2}{4} \big( \langle \delta x(t) \delta x(t) \delta x(t') \delta x(t') \rangle - \langle \delta x(t) \delta x(t) \rangle \langle \delta x(t') \delta x(t') \rangle \big) \right\}. \quad (34)$$

The integral of the first order term can be evaluated as

$$\int_0^{\tau_m} \int_0^{\tau_m} dt dt' \langle \delta x(t) \delta x(t') \rangle = \int_0^{\tau_m} \int_0^{\tau_m} dt dt' \int_{-\infty}^{\infty} d\omega e^{i\omega(t-t')} S_x(\omega)$$
$$= 4\tau_m^2 \int_{-\infty}^{\infty} d\omega \frac{\sin^2(\omega \tau_m/2)}{(\omega \tau_m)^2} S_x(\omega). \quad (35)$$

We again regularize the integral by introducing the low frequency cutoff $\omega_{\min}$, this yields a first order contribution to $\sigma_Y^2$ of

$$8 y_1^2 \alpha_x \int_{\omega_{\min} \tau_m}^{\infty} d\zeta \frac{\sin^2(\zeta/2)}{\zeta^3} \approx y_1^2 \alpha_x \big(3 - 2\gamma - 2\log(\omega_{\min} \tau_m)\big), \quad (36)$$

where for simplicity we used the limit $\omega_{\min} \tau_m \ll 1$.

The second term in (34) can be evaluated with the help of Wick's theorem which is valid given the assumption of the Gaussian noise. Specifically, we write

$$\int_0^{\tau_m} \int_0^{\tau_m} dt dt' \langle \delta x(t) \delta x(t) \delta x(t') \delta x(t') \rangle$$
$$= \int_0^{\tau_m} \int_0^{\tau_m} dt dt' \big\{ \langle \delta x(t) \delta x(t) \rangle \langle \delta x(t') \delta x(t') \rangle + 2 \langle \delta x(t) \delta x(t') \rangle \langle \delta x(t) \delta x(t') \rangle \big\}. \quad (37)$$

Note that the first term in (37) cancels with the last term in (34), while the second term in (37) can be written as

$$\int_0^{\tau_m} \int_0^{\tau_m} dt dt' \langle \delta x(t) \delta x(t') \rangle \langle \delta x(t) \delta x(t') \rangle$$
$$= \int_0^{\tau_m} \int_0^{\tau_m} dt dt' \int_{-\infty}^{\infty} \int_{-\infty}^{\infty} d\omega d\omega' e^{i(\omega+\omega')(t-t')} S_x(\omega) S_x(\omega')$$
$$= \tau_m^2 \int_{-\infty}^{\infty} \int_{-\infty}^{\infty} d\omega d\omega' S_x(\omega) S_x(\omega') \frac{\sin^2((\omega+\omega')\tau_m/2)}{((\omega+\omega')\tau_m/2)^2}. \quad (38)$$

Once again, we regularize the integral by introducing the low frequency cutoff $\omega_{\min}$ and get

$$2\tau_m^2 \alpha_x^2 \int_{\omega_{\min} \tau_m}^{\infty} \int_{\omega_{\min} \tau_m}^{\infty} d\zeta d\zeta' \frac{1}{\zeta \zeta'} \left( \frac{\sin^2((\zeta+\zeta')/2)}{((\zeta+\zeta')/2)^2} + \frac{\sin^2((\zeta-\zeta')/2)}{((\zeta-\zeta')/2)^2} \right). \quad (39)$$

The integral given above cannot be computed analytically for arbitrary values of $\omega_{\min}\tau_{\mathrm{m}}$. However, in the limit $\omega_{\min}\tau_{\mathrm{m}} \ll 1$ certain simplifications are possible. First, we perform one of the integrals in (39) and expand the result in powers of $\omega_{\min}\tau_{\mathrm{m}}$:

$$\int_{\omega_{\min}\tau_{\mathrm{m}}}^{\infty} d\zeta \frac{1}{\zeta\zeta'} \left( \frac{\sin^2((\zeta+\zeta')/2)}{((\zeta+\zeta')/2)^2} + \frac{\sin^2((\zeta-\zeta')/2)}{((\zeta-\zeta')/2)^2} \right) =$$
$$= \frac{4}{\zeta'^3} \left\{ -1 + (1+\gamma)\cos\zeta' - \mathrm{Ci}(\zeta') + \log(\zeta') + \zeta'\mathrm{Si}(\zeta') - \log(\omega_{\min}\tau_{\mathrm{m}})(1-\cos(\zeta')) \right\}$$
$$+ O(\omega_{\min}\tau_{\mathrm{m}}), \tag{40}$$

where $\mathrm{Ci}(\zeta') = -\int_{\zeta'}^{\infty} dt \cos(t)/t$ and $\mathrm{Si}(\zeta') = \int_0^{\zeta'} dt \sin(t)/t$. Next, we integrate (40) over $\zeta'$ and expand the result in powers of $\omega_{\min}\tau_{\mathrm{m}}$ once again. This yields the expression for the second order contribution to $\sigma_Y^2$ in the limit $\omega_{\min}\tau_{\mathrm{m}} \ll 1$:

$$y_2^2 \alpha_x^2 \int_{\omega_{\min}\tau_{\mathrm{m}}}^{\infty} \int_{\omega_{\min}\tau_{\mathrm{m}}}^{\infty} d\zeta d\zeta' \frac{1}{\zeta\zeta'} \left( \frac{\sin^2((\zeta+\zeta')/2)}{((\zeta+\zeta')/2)^2} + \frac{\sin^2((\zeta-\zeta')/2)}{((\zeta-\zeta')/2)^2} \right) \approx$$
$$\approx y_2^2 \alpha_x^2 \left( 7 - 6\gamma + 2\gamma^2 + (4\gamma-6)\log(\omega_{\min}\tau_{\mathrm{m}}) + 2\log^2(\omega_{\min}\tau_{\mathrm{m}}) \right). \tag{41}$$

Hence, the expression for variance in the limit $\omega_{\min}\tau_{\mathrm{m}} \ll 1$ takes the form

$$\sigma_Y^2 \approx y_1^2 \alpha_x \left( 3 - 2\gamma - 2\log(\omega_{\min}\tau_{\mathrm{m}}) \right)$$
$$+ y_2^2 \alpha_x^2 \left( 7 - 6\gamma + 2\gamma^2 + (4\gamma-6)\log(\omega_{\min}\tau_{\mathrm{m}}) + 2\log^2(\omega_{\min}\tau_{\mathrm{m}}) \right). \tag{42}$$

## E Temperature dependence of detuning noise

In this Appendix we briefly analyze temperature dependence of the detuning noise described in Section 4 of the main text. Fig. 8 illustrates signal (20) and noise (21) calculated as a function of system temperature for the average QD charge $(\mathcal{S}_n, \mathcal{N}_n)$ and the differential capacitance of the QD $(\mathcal{S}_C, \mathcal{N}_C)$. Both dependencies in Fig. 8 are plotted for values of detuning and phase corresponding to (or close to) the maximum visibility points, see discussion in Section 4 of the main text.

The signal strength in Fig. 8 decreases with temperature due to the fact that the energies (6) are symmetric with respect to $\varepsilon = 0$ line, see Fig. 2(a), and hence the difference between the observables for $p = +1$ and $p = -1$ vanishes at large $T$. The noise strength also usually decreases with temperature because the slope of the observables plotted as a function of $\Delta$ decreases with temperature too. However, there are certain parameter regimes when the slope is already in the saturation and hence it rises with $T$ thus increasing the noise strength as well, see for example Fig. 8(a) for $T/t \lesssim 1.5$. Overall, based on Fig. 8 we conclude that lowering $T$ benefits the measurement visibility.

## F Derivation of the expression for $C_{diff,-}$ in the limit $|\bar{t}_-| \ll T, \sigma_\Delta$

Expressions for energy and differential capacitance for the ground and exited states are given in Eqs. (6) and (8):

$$\varepsilon_p^{gr} = -\frac{1}{2}\sqrt{\Delta^2 + 4|\bar{t}_p|^2} = -\varepsilon_p^{exc}, \tag{43}$$

$$\frac{C_{\mathrm{diff},p}^{gr}}{C_g^2/C_{\Sigma,\mathrm{D}}} = -\frac{4\varepsilon_C|\bar{t}_p|^2}{(\Delta^2 + 4|\bar{t}_p|^2)^{3/2}} = -\frac{C_{\mathrm{diff},p}^{exc}}{C_g^2/C_{\Sigma,\mathrm{D}}}. \tag{44}$$

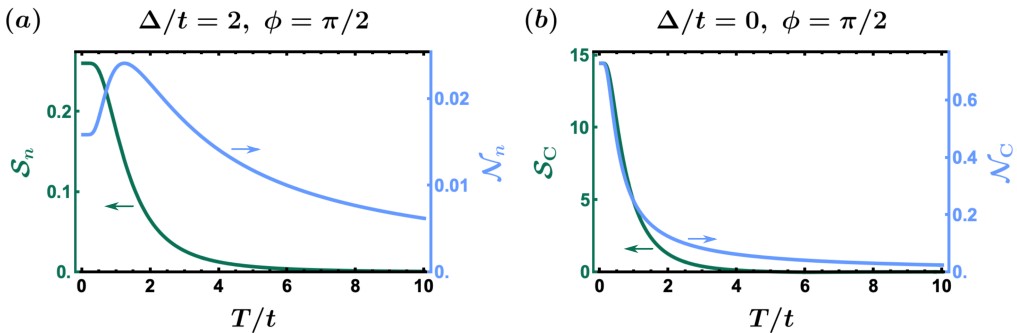

Figure 8: Signal (20) and noise (21) for the 2-MZM measurement of the average QD charge $\langle n_{\text{QD}} \rangle$(a) and differential QD capacitance $C_{\text{diff}}/C_{\Sigma,\text{D}}$(b) as a function of temperature $T$ for $\phi = \pi/2$ and different values of $\Delta/t$. Here we set $|t_1| = t$, $|t_2| = 1.5t$, $t = \varepsilon_C/5 = 0.02$ meV, $C_g/C_{\Sigma,\text{D}} = 2$ and the noise is detuning noise of strength $\sqrt{\alpha_C} = 0.01$.

In the high temperature limit temperature-averaged $C_{\text{diff},-}$ becomes

$$C_{\text{diff},-} = \frac{C_{\text{diff},-}^{gr} e^{-\varepsilon_-^{gr}/T} + C_{\text{diff},-}^{exc} e^{-\varepsilon_-^{exc}/T}}{e^{-\varepsilon_-^{gr}/T} + e^{-\varepsilon_-^{exc}/T}} \xrightarrow[T \gg |\bar{t}_-|, \Delta]{} -\frac{C_{\text{diff},-}^{exc} \varepsilon_-^{exc}}{T} \propto \frac{|\bar{t}_-|^2}{T(\Delta^2 + 4|\bar{t}_-|^2)}. \tag{45}$$

Given Gaussianly distributed random varaible $\Delta$ with variance $\sigma_\Delta^2$, averaging $C_{\text{diff},-}$ over the distribution gives in the limit $\sigma_\Delta \gg |\bar{t}_-|$

$$C_{\text{diff},-} \propto \frac{|\bar{t}_-|^2}{T \sigma_\Delta} \times \frac{1}{|\bar{t}_-|} = \frac{|\bar{t}_-|}{T \sigma_\Delta}. \tag{46}$$

## G   Noise in MZM-QD couplings

Noise in the MZM-QD coupling amplitudes $|t_1|, |t_2|$ results from the noise in electrostatic gates controlling those couplings. Similarly to the case of the detuning noise (22) in the main text we assume that the coupling noise has $1/f$ power spectrum:

$$S_t(\omega) = t^2 \frac{\alpha_t}{|\omega|}, \tag{47}$$

where we explicitly separated MZM-QD coupling energy $|t_1|, |t_2| \sim t$ and dimensionless noise strength $\alpha_t$.

We estimate $\alpha_t$ using experimental measurements of the dephasing time in gatemon qubits that use quantum wire suitable for topological superconductivity. Reference [51] reports $T_2^* \sim 4$ $\mu$s in InAs/Al based gatemon with qubit frequencies $f_Q \sim 5$GHz. We assume that $f_Q \propto \sqrt{E_J} \propto \sqrt{g_J}$, where $E_J, g_J$ are the Josephson energy and dimensionless conductance of the junction. We can then obtain an upper bound on the fluctuations of $g_J$ by assuming that the dephasing is dominated by noise in the dimensionless conductance of order $\Delta g_J$ which yields the estimate $\Delta g_J/g_J = 1/(\pi T_2^* f_Q) \sim 2 \times 10^{-5}$. This can be used to estimate the fluctuations in $t$ which is proportional to $\sqrt{g_J}$ of the junction connecting the qubit island and the QD. Assuming similar relative fluctuations of $g_J$ yields $\sqrt{\alpha_t} \sim 10^{-5}$. Note that the amount of variations in the conductance of a junction due to charge noise in the environment does depend on the regime in which the junction is operated. Junctions that are operated close to pinch off will likely show a stronger susceptibility to fluctuations. Nevertheless, the significantly smaller

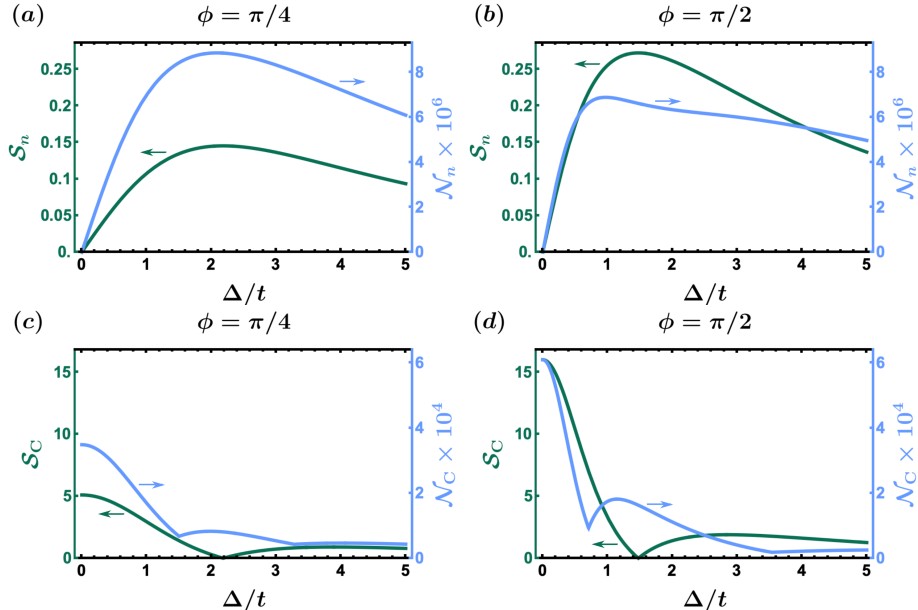

Figure 9: Effect of noise in the MZM-QD coupling. Signal (20) and noise (21) for the 2-MZM measurement of the average QD charge $\langle n_{\mathrm{QD}} \rangle$ (a)-(b) and differential QD capacitance $C_{\mathrm{diff}}/C_{\Sigma,\mathrm{D}}$ (c)-(d) as a function of detuning $\Delta$ for different values of $\phi$. Here we assume that the system is in its ground state ($T = 0$) and set $|t_1| = t$, $|t_2| = 1.5t$, $t = \varepsilon_{\mathrm{C}}/5 = 0.02$ meV, $C_g/C_{\Sigma,\mathrm{D}} = 2$, and the strength of the coupling noise $\sqrt{\alpha_t} \sim 10^{-5}$.

value of $\sqrt{\alpha_t} \ll \sqrt{\alpha_{\mathrm{C}}}$ obtained in the above estimate makes it unlikely that the noise in the tunnel coupling overcomes the detuning noise.

Similar to the case of the detuning noise in the main text we analyze effects of the coupling noise perturbatively. Using expressions for expectation value and variance of the observables (18),(19) we calculate signal and noise via Eqs. (20),(21). The perturbative treatment of the noise is well satisfied since $\sqrt{\alpha_t} \ll 1$. Fig. 9 illustrates the signal $\mathcal{S}_n$ ($\mathcal{S}_{\mathrm{C}}$) and the noise $\mathcal{N}_n$ ($\mathcal{N}_{\mathrm{C}}$) calculated for the average QD charge (a)-(b) (differential capacitance of the QD (c)-(d)) as a function of detuning. The signal lines in Fig. 9 resemble the ones in Fig. 6 and we refer reader to the main text for the discussion of the signal. Coupling noise, on the other hand, has a behavior qualitatively different from its detuning counterpart. First, in contrast to the detuning noise, the coupling noise $\mathcal{N}_n$ vanishes at zero detuning as illustrated in Fig. 9(a)-(b). This is associated with $\langle n_{\mathrm{QD},p} \rangle$ being identically zero at $\Delta = 0$ for any value of $|\bar{t}_p|$. At the same time, the local minimum at $\Delta = 0$ of the detuning noise $\mathcal{N}_{\mathrm{C}}$, see Fig. 6(c)-(d), is not present in the case of the coupling noise as can be observed in Fig. 9(c)-(d). The reason for this is that at $\Delta = 0$ the capacitance is affected by noise in the coupling already to first order as opposed of detuning noise which only acts at second order. Akin to the detuning noise, the effect of coupling noise vanishes together with the signal for large detuning values emphasizing that external noise sources, e.g. amplifier noise, would likely be dominant in that regime.

Overall, the SNR for the coupling noise exceeds $10^4$ in case of $\langle n_{\mathrm{QD}} \rangle$ and $C_{\mathrm{diff}}$ for most of the parameter values except vicinity of a few isolated points where the signal is fine tuned to zero. This signifies that due to $\sqrt{\alpha_t} \ll \sqrt{\alpha_{\mathrm{C}}}$ the MZM-QD coupling noise is not significant enough to affect the measurement visibility.

Intuitively, the weaker effect of noise on the tunnel coupling can be explained by differences in the sensitivity of the voltages controlling the tunnel coupling and the charge occupation. Assuming for simplicity a lever arm close to unity, the detuning changes significantly when the

corresponding voltage of the QD changes the charge occupation by one electron. This corresponds to voltages $\sim \varepsilon_C/e$ which is typically or the order of $0.1-1$mV. Changing the strength of the tunnel coupling on the other hand requires to sufficiently change the electrostatic potential in the tunneling barrier. The corresponding voltages are typically much larger $\sim 10-100$mV. Nevertheless, we caution that for sufficiently ill-behaved junctions which show sharp resonances in the dependency of the dimensionless conductance with respect to the junction gate voltage the general trend of weak coupling noise might be broken.

## H  Phase noise

The phase noise arises due to the noise in magnetic flux penetrating the enclosed area of the interference loop in the coupled island-QD setup, see Fig. 1. The flux noise, in turn, can originate from fluctuations in external magnetic field needed to tune the nanowires into the topological regime and/or from magnetic moments of electrons trapped in defect states of superconductors [52]. We estimate it by referring to the noise measurements in flux qubits which have interference loop based architecture similar to our topological setup. Refs. [53,54] observe $1/f$ behavior of the flux noise in flux qubits and report the noise value of $S_\Phi^{1/2}(1\text{ Hz}) \sim 1\mu\Phi_0\text{Hz}^{-1/2}$, where $\Phi_0 = h/(2e)$ is the superconducting flux quantum. Based on this we write the phase noise spectral power in our setup as

$$S_\phi(\omega) = \frac{\alpha_\phi}{|\omega|}, \tag{48}$$

with $\sqrt{\alpha_\phi} \sim 10^{-6}$.

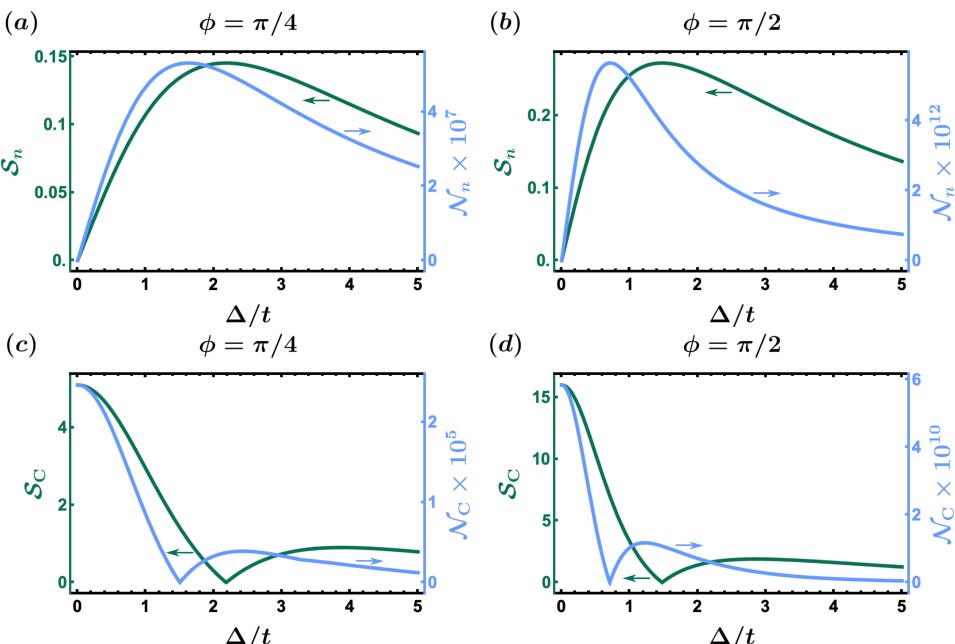

Figure 10: Effect of phase noise. Signal (20) and noise (21) for the 2-MZM measurement of the average QD charge $\langle n_{\text{QD}} \rangle$ (a)-(b) and differential QD capacitance $C_{\text{diff}}/C_{\Sigma,\text{D}}$ (c)-(d) as a function of detuning $\Delta$ for different values of $\phi$. Here we assume that the system is in its ground state ($T = 0$) and set $|t_1| = t$, $|t_2| = 1.5t$, $t = \varepsilon_C/5 = 0.02$ meV, $C_g/C_{\Sigma,\text{D}} = 2$, and the strength of the flux noise $\sqrt{\alpha_\phi} \sim 10^{-6}$.

Following analysis of the detuning noise in the main text and the MZM-QD coupling noise in Appendix G, here we treat $1/f$ phase noise perturbatively and calculate corresponding signal $\mathcal{S}$ and noise $\mathcal{N}$ via Eqs. (20),(21). Note that $\sqrt{\alpha_\phi} \ll 1$ is needed for the perturbative treatment of the noise to work. The results of the phase noise calculations are illustrated in Fig. 10(a)-(b) for the average QD charge ($\mathcal{S}_n, \mathcal{N}_n$) and Fig. 10(c)-(d) for the differential capacitance of the QD ($\mathcal{S}_C, \mathcal{N}_C$) as a function of detuning. The signal lines in Fig. 10 closely resemble the ones in Fig. 6 so the discussion of the signal can be found in the main text. On the other hand, the phase noise lines in Fig. 10 are qualitatively similar to the coupling noise lines in Fig. 9, see Appendix G for the corresponding discussion. The main difference between the phase noise and the coupling noise is that the phase noise is smaller: the SNR for the phase noise exceeds $10^5$ in case of both $\langle n_{QD} \rangle$ and $C_{diff}$ for most of the parameter values except vicinity of a few isolated points in parameter space where the signal is fine tuned to zero. Near $\phi = \pi/2$ the SNR is greater than $10^{10}$ for most of detuning values. Note also that the dependence on $\phi$ of the phase noise is much more significant than the dependence on $\phi$ of the coupling noise, cf. Fig. 10(a)-(b) and Fig. 9(a)-(b). Overall, we predict the phase noise to be not strong enough to affect the measurement visibility.

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
