# Peer review of "Visibility of noisy quantum dot-based measurements of Majorana qubits"

_SciPost Physics, doi:SciPost Phys. 10, 127 (2021)_

## Round 2 · Referee Report · Anonymous · 2020-11-5

Strengths

1. The paper is written in a very accessible manner and considers the problem of parity readout in Majorana qubits via quantum dots in a regime not studied in detail so far (namely for resonant dot-superconducting island cases).
2. Their study of noise effect is elegant and the results are of interest both for theorists and experimentalists alike.

Weaknesses

I see no significant weaknesses.

Report

The authors study the problem of parity readout of Majorana qubit via tunnel-coupled quantum dots (using the charge of the dot or the differential capitance) in the presence of (1/f) noise. The results are interesting and suggest that the resonant case (not considered in detail previously) may be of interest in practice. The results appear reasonable and correct, and are of interest to the communities working on Majorana physics or topological quantum computation. I have only a few minor comments (see below), and otherwise believe that the manuscript is of high quality and therefore can be published in SciPost.

Requested changes

1. eqs (9,10) should have only one equation number
2. I believe that in the introduction, when talking about the usefulness of Majorana qubits for linking Majorana detection and quantum computation, citation of PRL 124, 096801 (2020) is indicated.

  • validity: top
  • significance: high
  • originality: good
  • clarity: top
  • formatting: perfect
  • grammar: perfect

Author:  Aleksei Khindanov  on 2021-04-12  [id 1350]

(in reply to Report 1 on 2020-11-05)

We thank the referee for their report and implemented the requested changes in the resubmitted version.

---

## Round 2 · Referee Report · Anonymous · 2020-12-9

Strengths

1. addresses a timely problem of direct experimental relevance
2. employs an explicit noise model
3. uses concrete, experimentally motivated, parameter values

Weaknesses

1. presentation could be clearer

Report

The manuscript discusses the prospects of fermion parity measurement using a quantum dot based readout scheme, in particular the effects of various forms of intrinsic noise. This is a timely problem; predictions on how noise influences the visibility of measurement signals will be useful for upcoming Majorana experiments. I support the publication of the manuscript upon the authors addressing the questions and comments below and considering the requested changes.

Questions and comments:

1. The assumption of a single-level QD is stated as valid for high external magnetic fields for small enough QD. What is understood by 'high' and 'small enough' here? How does the requirement of 'high' magnetic field compare to the magnetic field tolerance of the superconducting ingredients involved?

2. The text, under Equation 4, discusses tuning the phase difference by varying the magnetic flux penetrating the enclosed area of the interference loop. Considering a superconductor + nanowire + applied magnetic field setup for Majorana fermions, to what extent does the alignment of the magnetic field along the nanowires (highlighted as a desired feature in [34]) pose challenges for tuning the flux in the interference loop?

3. The regime close to degeneracy is highlighted as beneficial for measurement visibility. At the same time, this is a regime characterized by the near degeneracy of certain charge configurations. Hence one might wonder: under what conditions on the parameters does the system continue to enjoy the charging energy induced protection against quasiparticle poisoning (now in the sense of the $N_\text{tot}$ and hence the combined parity $p$ being a good quantum number throughout the measurement process)? Which regimes should one avoid?

4. Equation 7 appears to arise by differentiation from Equation 2 (it would be helpful to state the origin of Equation 7 in the text). If so, it appears to involve an approximation neglecting the $\Delta$ dependence of the eigenvectors of $H_p$ in Equation 5. If such an approximation is indeed made: when is this approximation justified and when does it fail? Does it work near $\Delta=0$ and $\phi=\pi/2$?

5. The coupling-decoupling procedure and the extra charge measurement mentioned in the last paragraph of page 3 seem to be important ingredients of the fermion parity readout process. Could it be spelled out what can go wrong and how without these steps?

6. The charge noise is taken to be 1/f noise. Could the reader be reminded (and some references supplied) what justifies this form?

7. In the 3rd paragraph of Section III, could it be spelled out what is meant by "temperature average"?

8. In the penultimate sentence of the 3rd paragraph of Section III, scenarios are stated in which approach (1) vs approach (2) applies regarding the temperature. What is the reasoning/justification behind these statements?

9. In the paragraph under Equation 20, the high frequency cutoff is equated to the temperature. Why is this justified?

10. In the last paragraph of page 6, an analytic expression for the maximum charge signal is provided, and it is noted that for the authors' choice of parameters the noise term produces small corrections only. How sensitive is the match between the numerics and the formula to changing the choice of parameters?

11. The caption of Figure 5,6 (and the discussion in the paragraph below) refers to the noise at $T=0$. What is understood by this? Earlier, the text indicated that the temperature enters through the high frequency cutoff; if so then the $T\rightarrow0$ limit seems to involve divergences. The text (including Appendix E) would benefit from a clearer/more explicit discussion of how the temperature enters the considerations.

12. A related point: in the 2nd paragraph in page 7, what is the origin of the estimate $C_{\text{diff},-}\propto \bar{t}_-/(T\sigma_\Delta)$?

Requested changes

I believe that the text would benefit from updates addressing the questions and comments in the report. Furthermore, the authors might want to consider:

1. In the 2nd paragraph, defining the symbols $\sigma^z$, $\gamma_i$, $\gamma_j$.

2. Referring to Appendix E in the main text.

3. In the Introduction, penultimate paragraph: giving the gist of what is meant by "close to resonance" and "far-detuned". It becomes clear later on, but at this stage of the text it is hard to decipher.

4. In the 2nd paragraph of Sec. II B, rewording "single electron" to read "single excess electron" (or something similar).

5. Rewording the first sentence of the 3rd paragraph of page 4: it currently reads as if the coupling to $\gamma_1\ldots\gamma_4$ was exponentially small.

6. Clarifying, in the paragraph below Fig. 5, the meaning of "the differential capacitance of the smaller coupling". (The later parts of the texts suggest that this is the differential capacitance corresponding to the smaller of $|\bar{t}_+|$ and $|\bar{t}_-|$, but it would be useful to give the clear meaning at the point where the concept is introduced.)

7. In the second paragraph below Fig. 5, updating symbols $C_+$ and $n_+$ to the form in which they are defined (or defining them if they are meant to be new symbols).

8. In the Conclusion, refining the wording "...regime of small detunings which in general optimizes the SNR". (According to Section IV, the situation is not so clear cut: results include "From the perspective of pure charge noise the SNR would thus be largest for large detuning...".)

  • validity: high
  • significance: high
  • originality: good
  • clarity: good
  • formatting: excellent
  • grammar: good

Author:  Aleksei Khindanov  on 2021-04-12  [id 1351]

(in reply to Report 2 on 2020-12-09)
Category:
answer to question

We appreciate detailed feedback and addressed all the points raised by the referee in the point-by-point reply below. We also updated the manuscript where appropriate.

  1. The approximation of a single-level QD is valid when the energy difference between the two lowest levels of the dot is larger than the MZM-QD coupling. We updated the text to make this clear, see the paragraph after Eq.(2). In general the MZM-QD coupling is of the order $\sqrt{\delta \Delta_t}g$, where $\delta$, $\Delta_t$ and $g$ are the QD level spacing, topological gap and dimensionless conductance of the tunnel junction, respectively. Since the Zeeman field is required to close the trivial and reopen the topological gap it has to be larger than $\Delta_t$. Typical QD level spacing are also of similar size as $\Delta_t$ which makes the single-level approximation the most natural assumption.

  2. The magnetic field that drives the topological transition is indeed applied along the direction of the nanowire. The component of the field to tune the flux is much smaller and does not affect the properties of the topological wire. For a typical loop sizes of $1\mu\text{m}^2$ the required transversal field to tune the flux is of the order of a few mT. Any (small) out of plane component of the large applied magnetic field just provides a constant offset of the flux.

  3. The protection against external quasiparticle poisoning is still present even with the QD tuned to resonance. As pointed out in the manuscript the conserved quantity is now the combined charge of the QD-qubit system and their charging energies (or a galvanic disconnection) protects against external quasiparticles entering the system. As outlined at the end of Sec. IIA in the manuscript at some point one needs to map the combined parity to the qubit parity in the decoupling procedure which in principle can produce unwanted poisoning events. Such processes can be minimized by reestablishing the charging energy protection of the qubit from the QD by first detuning the QD and only then disconnecting the two.

  4. Equation 7 is derived by differentiation of Equation 6 as described by the first equality in Equation 7. As such, there are no approximations involved.

  5. Without these steps one will (with certain probability) infer an incorrect value of the Majorana parity $p_{12}$ during the measurement, thus reading out the qubit state incorrectly. This will in turn lead to a decrease in measurement and (in case of measurement-based topological quantum computing) gate fidelity. We updated the text to make this clear.

  6. $1/f$ charge noise has been observed in many solid-state quantum information processing devices, in particular superconducting and semiconducting qubits. See newly added Ref.[50] for a detailed review on the subject. Comparison with semiconducting qubits is particularly relevant for topological qubits considered in our study as both systems utilize semiconducting quantum dots at some point during quantum computation. Refs.[48,49] in the updated version of the draft analyze $1/f$ charge noise in semiconducting qubits.

  7. By temperature average we mean the expectation value with respect to an unperturbed (i.e. not subjected to noise) finite temperature density matrix. We updated the text to make this clear.

  8. We updated the discussion of how we incorporate temperature into the expectation values of observables, including the distinction between scenarios (1) and (2), which is hopefully now more clear. The expectation value $\langle \dots \rangle$ introduced below Eq.(17) is taken with respect to the environment for the noisy parameter. There are two opposing limits how to incorporate a finite temperature in the expectation values of the system operators. (1) The operator $\hat{Y}$ is temperature independent and the expectation value is taken with respect to the full density matrix of the system which includes both finite-temperature and noise effects; (2) the operator $\hat{Y}$ is already the temperature-averaged observable (i.e. the expectation value with respect to the unperturbed finite-temperature density matrix has been already taken) in which case taking the expectation value $\langle\dots\rangle$ amounts to only performing noise-averaging. For temperatures large enough that there is a significant occupation of the excited state, the timescales involved in the temperature fluctuations determine which of the two methods are more appropriate in capturing the variance of the measurement outcomes. If during the measurement time the system transitions frequently between the ground and excited state, the measurement will probe temperature averaged quantities (2), while for transitions slower than the measurement time the distribution of measurement outcomes would be broadened by temperature (1). To focus purely on the effect of the noise we take the limit (2) of long measurement times.

  9. We updated the discussion about the high frequency cutoff. Originally we wanted to use a high frequency cutoff originating from the physical processes that create the noise in the environment. We realize now that a more general argument for the high frequency cutoff can be obtained by considering the highest frequencies that the measurement apparatus can possibly resolve. Higher frequency fluctuations simply average out and are not detected as measurement noise. The measurements are performed by coupling resonators to the QD and observing shifts in the resonance frequency. This frequency thus provides a natural cutoff for the time scale the detector can resolve. Typical resonator frequencies are $\sim 1$GHz. Incidentally, this provides a very similar cut off as we used before since $1\text{GHz}$ corresponds to a temperature scale of $50$mK. The described change in the cutoff does not have a noticeable effect on the figures.

  10. The analytical formula for the detuning giving maximum charge signal is a good approximation as long as the noise-induced term in $\mathcal{S}_n$ is perturbative, i.e. much smaller then the noise-free term, and as such the choice of parameters should be corresponding. We updated the text to make it more clear.

  11. Temperature entering Figures 5-6 and 8-10 determines occupation of the excited state(s) of the system. There is essentially no differnce from using the small temperature of typical experiments and zero temperature which is why we considered the latter. This temperature enters in a very different way that the high-frequency cutoff. With the new choice of the high-frequency cutoff this possible confusion is resolved by not involving temperature in the cutoff choice.

  12. We added derivation of the expression for $C_{diff,-}$ to the Appendix F and refer the referee there.

We also implemented the requested changes in the text.

---

## Round 2 · Referee Report · Anonymous · 2020-12-26

Report

The authors consider the effects of $1/f$ noise on measurement outcomes in coupled systems of quantum dots and Majorana zero modes. This is an important question that needs to be answered before such systems can be used for quantum computing practically. The authors find that the signal-to-noise ratio is large and measurements are preferable when the detuning between the electron states constructed from pairs of Majorana modes is much smaller than the charging energy of the underlying superconductor-quantum dot system. In this regime, the low-energy physics is governed by the Majorana fermions, and is expected to be most relevant for Majorana-fermion-based quantum computing. The authors 2 geometries, consisting of 1 dot + 2 Majorana and 2 dots + 4 Majoranas, respectively, and consider various physics quantities relevant to these systems to arrive at and support the above general result.

Overall, I find the paper well-written and valid. Before discussing the effects of noise, they first describe the system and the measurements clearly, which I found helpful as I read the paper. The authors are also clear about the regime they have assumed and the limitations of the result (e.g. ignoring external sources of noise). My main suggestion to the authors, to strengthen the paper, would be to include a discussion of why the internal noise is mainly $1/f$. Are other noise profiles negligible? If so, why? Other than this, I find the paper suitable for publication.

  • validity: high
  • significance: good
  • originality: good
  • clarity: high
  • formatting: excellent
  • grammar: perfect

Author:  Aleksei Khindanov  on 2021-04-12  [id 1352]

(in reply to Report 3 on 2020-12-26)

We thank the referee for their review and the suggestion of including a discussion of why the internal noise is mainly $1/f$. We address this question below and updated the manuscript correspondingly (see list of changes).

$1/f$ charge noise has been observed in many solid-state quantum information processing devices, in particular superconducting and semiconducting qubits. See newly added Ref.[50] for a detailed review on the subject. Comparison with semiconducting qubits is particularly relevant for topological qubits considered in our study as both systems utilize semiconducting quantum dots at some point during quantum computation. Refs.[48,49] in the updated version of the draft analyze $1/f$ charge noise in semiconducting qubits.

---

## Round 3 · Author Response

Modifications and clarifications addressing the referee reports. See list of changes and referee replies for details.

---

## Round 3 · List of Changes

• Added reference [37, 48-50].
  • On page 1, bottom of the left column, added description of what is meant by $\sigma_z$ and $\gamma_{i/j}$.
  • On page 2, left column, second paragraph, added description of what is meant by "resonance" and "far-detuned regime".
  • On page 2, paragraph after Eq.(2), explained in which case the regime of a single-level QD is appropriate.
  • At the end of page 3, added a sentence describing implications of incorrect readout of the Majorana parity on quantum computation.
  • On page 4, first full paragraph after Eq.(9), changed "single electron" to "single excess electron".
  • On page 4, last paragraph in the left column, rephrased the first sentence such that it is more clear what is meant by "exponentially small" couplings.
  • On page 5, first paragraph in the right column, added a sentence which describes the motivation for using $1/f$ noise model.
  • At the end of page 5 and beginning of page 6, rewrote discussion on how we treat temperature when calculating expectation values.
  • On page 6, paragraph after Eq.(19), added physical motivation for the higher frequency cutoff $\tau_c^{-1}$.
  • On page 7, second paragraph in the left column, added reference to Appendix E.
  • On page 7, second to last paragraph in the left column, added that the noise-induced term in $S_n$ is perturbative and description of what is meant by that.
  • On page 7, first paragraph in the right column, described in more detail what is meant by "smaller coupling".
  • Throughout page 7, changed notation from $\bar{t}-\to |\bar{t}-|$, $\bar{t}+\to |\bar{t}+|$, $\langle C_+\rangle\to C_{\text{diff},+}$ and $\langle n_+ \rangle\to \langle n_{\text{QD},+} \rangle$.
  • Added Appendix F.
  • On page 7, second paragraph in the right column, added reference to Appendix F.
  • On page 8, first paragraph in the right column, clarified that the regime of small detunings optimizes the SNR only in the presence of external noise.

---

## Editorial Decision

published